# A Comprehensive Review of COVID-19-Related Olfactory Deficiency: Unraveling Associations with Neurocognitive Disorders and Magnetic Resonance Imaging Findings

**DOI:** 10.3390/diagnostics14040359

**Published:** 2024-02-07

**Authors:** Ludovica Simonini, Francesca Frijia, Lamia Ait Ali, Ilenia Foffa, Cecilia Vecoli, Carmelo De Gori, Sara De Cori, Monica Baroni, Giovanni Donato Aquaro, Carlo Maremmani, Francesco Lombardo

**Affiliations:** 1Institute of Clinical Physiology, National Research Council (CNR), 54100 Massa, Italy; ilenia.foffa@cnr.it (I.F.); cecilia.vecoli@cnr.it (C.V.); 2Bioengineering Unit, Fondazione Toscana G. Monasterio, 56124 Pisa, Italy; ffrijia@ftgm.it; 3Pediatric Cardiology and GUCH Unit, Fondazione “G. Monasterio” CNR-Regione Toscana, 54100 Massa, Italy; 4Department of Radiology, Fondazione Monasterio/CNR, 56124 Pisa, Italy; cdegori@ftgm.it (C.D.G.); sdecori@ftgm.it (S.D.C.); lombardo@ftgm.it (F.L.); 5Fondazione “G. Monasterio” CNR-Regione Toscana, 54100 Massa, Italy; baroni@ftgm.it; 6Academic Radiology Unit, Department of Surgical, Medical and Molecular Pathology and Critical Area, University of Pisa, 56124 Pisa, Italy; giovanni.aquaro@unipi.it; 7Unit of Neurology, Ospedale Apuane, Azienda USL Toscana Nord Ovest, 54100 Massa, Italy; carlo.maremmani@uslnordovest.toscana.it

**Keywords:** COVID-19, MRI, olfactory, imaging, neurocognitive, mood disorders, anosmia, olfactory training

## Abstract

Olfactory dysfunction (OD) is one of the most common symptoms in COVID-19 patients and can impact patients’ lives significantly. The aim of this review was to investigate the multifaceted impact of COVID-19 on the olfactory system and to provide an overview of magnetic resonance (MRI) findings and neurocognitive disorders in patients with COVID-19-related OD. Extensive searches were conducted across PubMed, Scopus, and Google Scholar until 5 December 2023. The included articles were 12 observational studies and 1 case report that assess structural changes in olfactory structures, highlighted through MRI, and 10 studies correlating the loss of smell with neurocognitive disorders or mood disorders in COVID-19 patients. MRI findings consistently indicate volumetric abnormalities, altered signal intensity of olfactory bulbs (OBs), and anomalies in the olfactory cortex among COVID-19 patients with persistent OD. The correlation between OD and neurocognitive deficits reveals associations with cognitive impairment, memory deficits, and persistent depressive symptoms. Treatment approaches, including olfactory training and pharmacological interventions, are discussed, emphasizing the need for sustained therapeutic interventions. This review points out several limitations in the current literature while exploring the intricate effects of COVID-19 on OD and its connection to cognitive deficits and mood disorders. The lack of objective olfactory measurements in some studies and potential validity issues in self-reports emphasize the need for cautious interpretation. Our research highlights the critical need for extensive studies with larger samples, proper controls, and objective measurements to deepen our understanding of COVID-19’s long-term effects on neurological and olfactory dysfunctions.

## 1. Introduction

SARS-CoV-2 is the virus responsible for the COVID-19 pandemic, which, as of August 2023, has caused nearly 7 million deaths worldwide, according to World Health Organization (WHO) data [1]. COVID-19 can present with a wide range of respiratory and extra-respiratory symptoms, including a broad spectrum of acute neurological symptoms involving both the central and peripheral nervous systems, such as headaches, dizziness, olfactory and gustatory deficits, polyneuropathies, and even encephalitis or strokes. Some studies have highlighted how neurological sequelae are present in a third of patients in the first six months following acute COVID-19 infection [2,3,4].

Within the context of a COVID-19 infection, acute olfactory dysfunction (OD) is characterized by a diminished or altered sense of smell lasting 14 days or less. This definition specifically excludes cases involving chronic rhinosinusitis, a history of head trauma, or the use of neurotoxic medications [5].

Since the beginning of the pandemic, the loss or alteration of the sense of smell and taste has been reported as among the most common symptoms, with an incidence approximately 10 and 9 times higher than in other respiratory infections, respectively. Olfactory and gustative dysfunctions can manifest in various ways among COVID-19 patients, ranging from a complete loss of perception (anosmia and ageusia) to reduced perception (hyposmia and hypogeusia) and even distorted perception (parosmia and parageusia). In some cases, individuals may experience sensations without external stimuli, referred to as phantosmia and phantogeusia [2,6,7].

The olfactory deficit can also result from viral infections caused by other pathogens and traumas or be secondary to sinus diseases. Furthermore, it can also be a part of neurodegenerative pathological processes [8,9].

Olfactory dysfunction has shown an incidence of 30% to 75% among COVID-19 patients and exhibits a slight predominance in females [10].

It often proves to be the sole symptom in otherwise asymptomatic patients and is one of the earliest symptoms to appear before others develop. Typically, olfactory dysfunction arises around the third day after infection and usually resolves completely within 4–6 weeks. However, after 4 months, these alterations can persist in 27% of cases [11] and in 21.3% for up to a year [12]. The inability or impairment of the olfactory process can lead to a markedly reduced quality of life. The lack of pleasure in eating and drinking can lead to alterations in eating and social behaviors [13].

Despite the high prevalence of COVID-19-related OD, data on MR imaging findings are limited. However, neuroimaging abnormalities related to COVID-19-associated OD are gaining attention, especially abnormalities of the olfactory bulb (OB), olfactory sulcus (OS), olfactory cleft, and olfactory tract (OT). These imaging results would be useful to shed light on the still unclear mechanisms underlying the olfactory disorders associated with SARS-CoV-2 disease, offering more information on the mechanisms of virus entry and the involvement of anatomical structures. With the aim of clarifying the radiological findings of persistent COVID-19-related OD, we performed a literature review focusing on OB changes in patients with clinically confirmed post-COVID-19 OD. 

In addition, our team has undertaken an in-depth investigation to explore the connection between olfactory deficits caused by COVID-19 and the emergence of neurocognitive disorders. Through a detailed study, we sought to shed light on this complex correlation, considering the multiple facets involved in the pathology. Furthermore, within this overview, we have also examined current treatments under investigation for olfactory deficits resulting from COVID-19 to provide a clear perspective on this crucial subject, thereby contributing to a more profound understanding of the long-term consequences of COVID-19 on neurocognitive health.

## 2. Search Strategy

For this literature review, searches on online databases (including PubMed, Scopus, and Google Scholar) were carried out until 5 December 2023, using keywords such as “COVID-19”, “olfactory deficit”, “anosmia”, “imaging”, “MRI”, “olfactory bulbs”, “neurocognitive deficits”, “mood disorders”, “neuropsychiatric sequelae”, “treatments”, and “SARS-CoV-2”, applying various combinations of these keywords to each of the online databases. No restrictions were applied regarding publication date and country of publication to maximize the retrieval of relevant published studies. The included articles are 12 observational studies and 1 case report that assess structural changes in olfactory structures, highlighted through MRI, and studies correlating the loss of smell with neurocognitive disorders or mood disorders in COVID-19 patients, compared to control subjects, where possible. The review also encompasses articles on available and investigational treatments for olfactory loss caused by SARS-CoV-2. Additional articles related to this field were also included if appropriate.

## 3. Results

### 3.1. The Structure of the Olfactory System and Pathogenesis of Olfactory Disorders in COVID-19

#### 3.1.1. The Structure of the Olfactory System

The olfactory system is responsible for perceiving volatile chemical substances and gases present in the air (odorants) and is phylogenetically one of the most primitive sensory systems (Figure 1) [14,15]. The olfactory organ consists of the olfactory epithelium (OE), OB, OT, and the piriform olfactory cortex [14]. The transformation of olfactory signals into electrical signals occurs at the level of the olfactory epithelium that lines the apical region of the central mucosa, where three main types of cells can be distinguished: olfactory sensory neurons, responsible for transducing odor stimuli; support cells, similar to glial cells, which protect the neurons and help produce the mucus in which odor molecules are trapped; and basal cells, which are the progenitor stem cells of olfactory neurons [16]. The long and thin dendrite of olfactory neurons is equipped with long olfactory cilia, which are nonmotile but extend into the mucus layer and possess membrane olfactory receptors to which dissolved odor molecules bind [17]. The axons of the OSNs pass through the cribriform plate of the ethmoid bone to reach the OBs, where they form synaptic glomeruli. In the glomeruli of the OB, communication occurs between afferent olfactory neurons and second-order neurons, which transmit signals to the primary olfactory cortex [14]. The olfactory system uses a combinatorial receptor coding scheme to discriminate and identify odor molecules [18]. Further processing of olfactory stimulation takes place in higher cortical regions. The orbitofrontal cortex, amygdala, and hippocampus participate in the translation of olfactory stimulation and its incorporation into thought processes, emotional reaction perception, memory formation, and learning processes. It is believed that the multiregional involvement of olfactory stimuli, consisting of the amygdala, prepyriform cortex, entorhinal cortex, and hippocampus, explains the relationship between odor, emotions, and memory formation [14,19].

#### 3.1.2. Pathogenesis of Olfactory Disorders in COVID-19

Due to its high affinity for the central nervous system, SARS-CoV-2 is considered neurotropic, neuroinvasive, and neurovirulent, but some variants show a greater capacity than others. The ancestral variant D614G stands out, followed by the Gamma variant, Delta variant (B.1.617.2), and finally, Omicron BA1 (B.1.1.529). The latter shows lower neurotropism in in vitro and in vivo studies [10]. In a study conducted on the golden hamster, a relevant model for studying the pathogenesis of SARS-CoV-2 infection, 62.5% of animals infected with SARS-CoV-2 Wuhan showed loss of smell. Only 12.5% of animals infected with the Gamma variant had completely lost their sense of smell, while 62.5% exhibited compromised olfactory performance. In contrast, none of the animals infected with Delta and Omicron/BA.1 showed signs of olfactory compromise [6]. A recent study compared the frequency of self-reported loss of smell in 63.002 symptomatic COVID-19 subjects exposed during the peak prevalence period of Delta (from 1 June to 27 November 2021) with that of the same number of COVID-19 patients exposed during the peak prevalence period of Omicron (from 20 December 2021 to 17 January 2022). The two groups were grouped by gender, age, and vaccination dose. Loss of smell was less common in participants infected during Omicron compared to the peak Delta prevalence periods (16.7 vs. 52.7%, OR: 0–17; 95% CI: 0-16-0-19, *p* < 0.001) [20]. Currently, the exact pathogenesis and molecular mechanisms of OD in COVID-19 are unknown. Nevertheless, the impact of SARS-CoV-2 infection on the brain is undeniable. However, several hypotheses have been made to explain the emergence of OD in COVID-19 patients. One initial hypothesis suggested conductive hypo/anosmia, which arises from mechanical obstruction and is related to congestion and rhinitis. This can influence airflow and compromise the transport of odorants, causing olfactory loss, despite an intact OE [21]. However, this hypothesis can presumably be ruled out since several studies have reported that OD has a higher prevalence than nasal congestion in COVID-19 patients and that loss of smell often outlasts the duration of respiratory symptoms. This suggests that mechanisms other than sinonasal obstruction may play a role [22].

##### Direct Damage to Olfactory Neurons

Neurosensorial hypo/anosmia occurs due to direct damage to olfactory neurons and the OB by the virus. The hypothesis that this could be the cause of hypo/anosmia derived from COVID-19 has been questioned by numerous studies that have highlighted the absence of angiotensin-converting enzyme 2 (ACE2) and transmembrane serine protease 2 (TMPRSS2), the key factors for the virus’s entry into the cell [23,24]. These results have also been supported by the study conducted by Bryche et al. [25], which demonstrated that SARS-CoV-2 was not detected in the olfactory neurons of hamsters, suggesting that, since olfactory neurons do not express ACE2 and TMPRSS2, the virus should use another pathway to infect the olfactory system. However, in addition to being able to enter the OBs and affect the brain through transcribriform or vascular pathways [26], it has been shown that there are other receptors in the brain in addition to angiotensin-converting enzyme 2 (ACE2) that can facilitate the neuroinvasion of SARS-CoV-2, such as Basigin (BSG), neuropilin-1 (NRP1), transmembrane serine protease 11A (TMPRSS11A), and furin receptors, but their quantity varies from individual to individual [27,28,29]. 

##### Damage to the OE

Another mechanism that could explain the olfactory deficit is damage to the OE. Numerous studies have established the expression of ACE2 and TMPRSS in various support cells of the OE, namely Bowman’s gland cells, horizontal basal cells, pericytes of the OB, mitral cells, sustentacular cells, and microvillar cells [30]. Of these support cells, sustentacular cells have gained immense attention as the initial site of SARS-CoV-2 infection in the OE. In addition to their higher expression of ACE2 and TMPRSS2 compared to others, sustentacular cells are located on the surface of the nasal cavity, making them vulnerable to exposure to the external environment [31]. The loss of support cells and stem cells leads to thinning and lack of repair of the OE, resulting in the loss of olfactory dendrites, which likely explains prolonged OD [22,32]. 

##### Inflammation

Furthermore, persistent inflammation after invasion, resulting from the production of proinflammatory cytokines such as interleukin-6 (IL-6), tumor necrosis factor alpha (TNF-α), interferon-gamma (IFN-γ), and CXC chemokine ligand 10 (CXCL10), appears to contribute to the development of post-COVID-19 OD, as it can directly damage the OE or interfere with cell signaling processes [13,33]. 

In conclusion, a comprehensive understanding of the pathogenetic mechanisms of OD following SARS-CoV-2 infection requires further research to precisely delineate the entry pathways of the virus into the olfactory system and to fully comprehend the roles of the various receptors involved. A detailed overview of the underlying pathogenic mechanisms of OD associated with SARS-CoV-2 infection is illustrated in Figure 2. 

### 3.2. Magnetic Resonance Imaging Olfactory System Findings 

#### 3.2.1. Correlation between Olfactory System MRI Findings and OD in COVID-19 Patients

Magnetic resonance imaging (MRI) is the gold standard in the etiological assessment of persistent OD (>2 months) and could be indicated after a SARS-CoV-2 infection [34]. Descriptions of MRI findings from all available studies included the results for morphological and volumetric abnormalities and/or increased signal intensity of the OBs, alterations in the depth of OS, and anomalies of the olfactory cortex, as well as irregularities of neuron filia, that were reported in patients with OD after COVID-19 infection. In the majority of the selected papers, OD was evaluated objectively with the Sniffin’ Stick Test, the abnormal European Test of Olfactory Capabilities (ETOC), and the University of Pennsylvania Smell Identification Test (UPSIT). Each study used different sequences for the evaluation of MRI parameters, which are summarized in Table 1. Few imaging studies incorporate advanced techniques like olfactory fMRI that could be helpful to better understand the underlying pathogenesis, and guide patient management for cases with persistent OD [35,36]. fMRI is a method that could evaluate alterations in the olfactory structure of COVID-19 patients suffering from OD. Yildirim et al. [35]. suggested that there was no significant difference in orbitofrontal and entorhinal cortex activity between COVID-19-related OD and other post-infectious OD, whereas trigeminal sensory activity was more robust in COVID-19-related OD. These findings may reflect a better preserved central olfactory system in COVID-19-related OD compared to other post-infectious OD. Iravani et al. [36], instead, with an fMRI analysis, revealed significantly decreased activity in the superior frontal lobe and basal ganglia among patients with olfactory OD when compared to the control group. The purpose of the reported studies was to determine if there is imaging evidence of olfactory apparatus pathology in patients with COVID-19 and neurologic symptoms. To reach this purpose, a dedicated MRI study allows the assessment of OB morphology, volume (OBV), and signal intensity, status of the olfactory nerve filia, and signal intensity of the primary olfactory cortex [35,36].

#### 3.2.2. OBV and OS Depth Changes

The most commonly measured parameters for the evaluation of the olfactory system are OBV and OS depth. OBV was calculated as a summation of manually drawn sequential regions of interest (ROIs) on consecutive coronal T2 sequences. OS depth represents the distance between the deepest point of the OS and a line tangent to the inferior borders of the rectus and medial orbital gyri. A recent paper [37] reported a decrease in OBV and OS depth for both the right and left side in hospitalized COVID-19 patients (*n* = 31) with anosmia and hyposmia compared with a healthy group (*n* = 35, age-matched), which supports direct damage to olfactory neuronal pathways that return back to normal function. A decrease in OBV and OS depths on MRI was shown by Gore et al. [38] who aimed to evaluate if there may be a link between cardiac arrhythmias and olfactory anatomical abnormalities. The study included 44 patients with cardiac arrhythmias compared to 43 healthy control patients, and 11 patients with acute COVID-19 were also compared in those groups. Both cardiac and COVID-19 patients showed a reduction in OBV and OS depth. The first limit of the study was that the patients were significantly older than the controls, and the second one is that with a multivariate analysis, the significant difference remains for smaller OS depth but not for smaller OBV [38]. The reduction in OBV was reported by a group in two different papers [35,39]. In particular, a paper evaluated 23 patients with persistent COVID-19 OD [39], and a second paper followed 31 COVID-19 patients with persistent OD and 97 patients with other post-infectious OD (after upper respiratory tract infection) [35]. Both papers showed a decreased OBV. In particular, the decrease in OBV appears not as pronounced as in other post-infectious OD [35]. For the OS depth, no significant difference was found between the two groups for Yildirim et al. [35]. Capelli et al. [2] also showed significantly lower left, right, and total OBV in COVID-19 patients (196 patients of which 78 reported olfactory loss as the only neurological symptom) compared to 39 controls. They processed MRI images by ad hoc semi-automatic processing procedures. The OBV was measured on T2-weighted MRI based on manual tracing and normalized to the brain volume. Also, the results of Seleim et al. [40] who studied anosmic patients with paranasal sinus CT and MRI were consistent with most of the previous studies reported. In fact, they found significant decreases in the values of both OBV and OS depth. Instead, Abdou et al. [41], in a study that compared 110 patients with post-COVID-19 OD and a control group of 50 normal subjects, showed significantly increased OB dimensions (length × width × height) and volume compared with controls. They hypothesize that the mechanism underlying COVID-related OD is sensorineural loss through virus spread and damage to the OE and pathways. The apparent contrast result indicates a transient inflammation of OB that occurred immediately after a negative nasopharyngeal swab, which is reversible in most patients in a follow-up MRI examination [42,43]. Brudasca et al. [34] evaluated the relationship between the OD severity (anosmia, severe hyposmia, moderate hyposmia, and mild hyposmia) and OBV measured by MRI and demonstrated no significant differences in terms of visual analysis and OBV measurement between the groups of patients considered. 

#### 3.2.3. OBs Signal Intensity 

The normal imaging appearance of the OB is well described in morphology and demonstrates uniform T2 signal intensity. Strauss et al. [4] compared the OB and OT signal intensity, on thin-section T2 WI and postcontrast 3D T2 FLAIR images, in 12 patients with COVID-19 and neurological symptoms (one with anosmia) and a group of 12 patients with OD that was not COVID-19 related, demonstrating a significant difference in OB signal intensity between the two groups (greater in the patients with COVID-19 and neurologic symptoms). Kandermirli et al. [39] highlighted that 91.3% of cases with persistent COVID-19-related OD had abnormalities in the form of a diffusely increased OB signal intensity. Capelli et al. [2] quantified, on 3D T2 FLAIR sequences, the OT median signal intensity, but no significant evidence was found in the COVID-19 patient group compared with control subjects, except for a few outliers. Li et al. [44], in a case report, reported a 21-year-old male who had presented with a loss of smell for five days without any respiratory problems or fever and revealed hyperintensities inside bilateral olfactory nerves, suggestive of bilateral olfactory neuropathy beyond a smaller right OB.

#### 3.2.4. OBs Morphological Abnormalities 

The shape of the OB morphology was assessed on coronal T2 images. OBs normally have an oval or inverted-J shape. Kandemirli et al. [39], in a small group (*n* = 23) of COVID-19 patients with persistent anosmia, reported a change in OB shape (normal shape *n* = 8; mild irregularity with preserved J shape *n* = 2; contour lobulations *n* = 5; and rectangular shape *n* = 8); the same team, in another paper [35], compared COVID-19 patients with persistent OD and patients with other post-infectious OD and demonstrated no significant difference in the two groups.

#### 3.2.5. Olfactory Neuron Filia Abnormalities 

Normally olfactory nerve filia show a fine architecture with uniform distribution of the filia at regular intervals. Focal thickening of the filia with nonuniform distribution was considered abnormal clumping. Yildirim et al. [35] demonstrated that there was a significantly higher rate of olfactory nerve clumping in COVID-19-related OD than post-infectious OD. Furthermore, Kandermirli et al. [39] reported an evident clumping of olfactory filia (in 34.8% of cases studied in the paper; thinning with scarcity of filia was reported in 17.4% of the cases). 

#### 3.2.6. Olfactory Cortex and White Matter Olfactory Region Abnormalities 

Different papers reported structural brain changes in patients with persistent OD after coronavirus disease [35,37,45,46]. In post-COVID-19 patients with persistent OD, Yildirim et al. [35] used diffusion tensor imaging (DTI) data to study white matter (WM) integrity. The authors found that the white matter tract integrity of olfactory regions was better preserved in COVID-19 anosmia compared to other post-infectious OD [35]. A recent work investigates whether COVID-19 patients with prolonged OD have structural brain changes compared to COVID-19 patients without OD [45]. The authors demonstrated with a voxel-based morphometry analysis that GM volume (in the involved regions: left amygdala, insular cortex, parahippocampal gyrus, frontal superior and inferior orbital gyri, gyrus rectus, olfactory cortex, caudate, and putamen) decreases in COVID-19 patients with OD compared to patients without OD; in the DTI analyses, MD increased in the olfactory system. These findings might explain why some COVID-19 patients have not recovered their sense of smell [45]. The chronic effects of COVID-19 on gray matter were investigated by Perlaki et al. [46], measuring the cortical thickness and subcortical volume on the 3D T1 images using Freesurfer 6.0 image analysis software in a group of 38 patients who recovered from mild COVID infection without a history of clinical long COVID and 37 healthy control subjects. Parlak [37] showed significantly lower bilateral mean global cortical thickness, lower subcortical gray matter, and lower right OBV in COVID-19 patients.

**Table 1 diagnostics-14-00359-t001:** This table provides a consolidated overview of MRI studies, integrating details on scanner specifications and imaging sequences with participant information, and key findings. This comprehensive table serves as a valuable reference for understanding the technical methodologies employed across various investigations and their corresponding clinical outcomes.

Authors	Scanner Field	Sequence	Patients	Controls	Findings
Strauss S.B. et al. (2020) [4]	3T (Signa Architect and Discovery 750W, GE Healthcare Waukesha, WI, USA) 3T (Skyra, Siemens Healthineers, Erlangen, Germany)	2D Cor T2-WI 3D T2 FLAIR	Patients with COVID-19 and neurological symptoms (*n* = 12 including 1 with OD)	Patients with non-COVID-19 OD (*n* = 12)	Increase signal intensity in OB
Kandemirli S.D. et al. (2021) [39]	3T (Magnetom, SiemensHealthineers, Erlangen, Germany)	2D Cor T2-WI Ultra-high-resolution T2-SPACE	Patients with persistent COVID-19-related OD (*n* = 23)	NO	Decrease in OBV; Abnormality in OB signal intensity; Clumping of olfactory filia
Yildirim D. et al. (2021) [35]	3T (Magnetom, Siemens Healthineers, Erlangen, Germany)	2D Cor T2-WI 3D T2 FLAIR Ultra-high-resolution T2-SPACE DTI fMRI (EPI)	Patients with persistent related COVID-19 OD (*n* = 31)	Patients with post-infectious OD other than COVID-19 (*n* = 97)	Decrease in OBV; Increased OB signal intensity; Clumping of olfactory filia
Li et al. (2021) [44]	Not reported	2D Cor T2-WI 3D TSE	Patient with COVID-19 and OD (*n* = 1)	NO	Decrease in right OBV; Hyperintensities in bilateral olfactory nerves
Brudasca I. (2022) [34]	Not reported	2D Cor T2-WI	Patients with related COVID-19 OD (*n* = 67)	NO	No OBV significant differences in between subgroups with mild, moderate, or severe hyposmia
Gore M.R. (2022) [38]	Not reported	3D T2-WI	Patients with cardiac arrhythmia (*n* = 44), Patients with COVID-19 (*n* = 11)	Healthy control (*n* = 43)	Decrease in OBV and OS depths
Abdou E.H.E. et al. (2023) [41]	1.5T (Ingenia, Philips Medical Systems, Eindhoven, The Netherland)	3D T1 FLAIR	Patients with related COVID-19 OD (*n* = 110)	Healthy control (*n*= 50)	Increased OBV and OB dimensions (length × width × height)
Capelli S. (2023) [2]	3T (Discovery 750W, GE Healthcare Waukesha, WI, USA)	2D Cor T2-WI 2D Sag T2 FLAIR 3D T2 FLAIR	Patients with COVID-19 (*n* = 196, *n* = 78 of them with OD)	Healthy control (*n*= 39)	Decrease in OBV; No significant differences in OT signal intensity
Iravani K. et al. (2023) [36]	1.5T (Magnetom Avento, SiemensHealthineers, Erlangen, Germany)	Ultra-high-resolution T2-SPACE fMRI (EPI)	Patients with related COVID-19 OD (*n* = 15)	Healthy control (*n*= 5)	Decreased activity in the superior frontal lobe and basal ganglia
Parlak A.E. et al. (2023) [37]	3T (Ingenia, Philips Medical Systems, Eindhoven, The Netherland)	2D Ax, Sag T2-WI 3D T1-WI	Patients with COVID-19, anosmia, and hyposmia (*n* = 31)	Healthy control (*n*= 35)	Decrease in OBV and OS depth
Perlaki G. (2023) [46]	3T (Magnetom PrismaFit, SiemensHealthineers, Erlangen, Germany)	2D Cor T2-WI 3D T1-WI	Patients who recovered from mild COVID infection (*n* = 38)	Healthy control (*n* = 37)	Lower bilateral mean cortical thickness, lower subcortical gray matter, and lower right OBV
Seleim A.M.A (2023) [40]	1.5T (Achieva, Philips Medical Systems, Eindhoven, The Netherland)	Not reported	Patients with related COVID-19 anosmia (*n* = 20)	NO	Decreases in OBV and OS depth
Campabadal A. (2023) [45]	3T (Magnetom Prisma, Siemens Healthineers, Erlangen, Germany)	2D SAG T1-WI 2D AX FLAIR DTI	Patient with COVID-19 and OD (*n* = 23)	Patient with COVID-19 without OD (*n* = 25)	Decrease in GM volume (areas reported in the main text); Increase MD in olfactory system

### 3.3. Correlation between OD and Neurocognitive Deficits

Up to a third of those recovered from COVID-19 complain of a broad spectrum of acute and chronic neurological disorders, such as cognitive deficit, “brain fog”, insomnia, headache, depression, anxiety, and mental fatigue, which interfere with full functional recovery (Figure 3) [47,48]. Among the cognitive disorders in the post-COVID phase, patients usually reported mental slowness, difficulties in paying attention, finding words, memory deficits, and planning daily activities [47,48,49]. Consequently, considering OD as an early biomarker of cognitive dysfunction in neurodegenerative diseases may also indicate the severe cognitive consequences of prolonged OD in COVID-19-infected patients [50,51,52,53]. Indeed, ODs have long been associated with mental disorders, but this connection has been further emphasized by the COVID-19 pandemic [51]. Mood disorders, such as anxiety, depression, and post-traumatic stress disorder, are also highly prevalent in the months following acute infection, especially among hospitalized patients. Anxiety disorders and depressive disorders are diagnosed in 11–19.5% and 13–15.9% of patients, respectively, following acute COVID-19 infection [3,54,55]. Different brain areas associated with emotion processing and cognitive functions overlap with the olfactory pathway [56,57]. Although the underlying pathogenesis of the relationship between OD and neurological deficits is still unclear, various processes may be involved, such as inflammation, alterations in the neurogenesis of peripheral and central structures of the olfactory system, and functional changes in brain structures [58,59,60]. Therefore, the link between the loss of smell and neurological deficits could be attributed to the effects of COVID-19 on common anatomical structures [56,57,61].

#### 3.3.1. Cognitive Impairment

Several studies have examined post-COVID cognitive impairments through global cognitive screening tests or more detailed neuropsychological assessments, revealing variable numbers of patients with impairment in one or more cognitive domains [3,55,61,62,63,64,65,66]. In a prospective study by Muccioli et al. [62], 23 individuals with persistent COVID-19-related OD and 26 age- and sex-matched healthy controls underwent olfactory and neuropsychological assessments. Memory and executive functions were the most affected cognitive domains, with 9% and 13% impairment in short-term and long-term verbal memory. Significant correlations were found between odor discrimination and executive functions. Anosmia emerged as a reliable predictor of mnemonic performance, aligning with findings in other studies. The study by Ruggeri et al. [55] demonstrated a positive correlation between transient anosmia during COVID-19 and memory impairments. Similar results were found in the study by Llana et al. [63] involving 60 patients, where participants reporting OD only during the acute phase of the disease scored lower on the Montreal Cognitive Assessment (MoCA) in overall cognition compared to those without ODs. In the study by Cecchetti et al. [64], 53% of patients exhibited impairments in at least one cognitive domain two months after COVID-19 resolution. At a 10-month follow-up, patients who reported hyposmia during the acute illness exhibited significantly less improvement in verbal memory tests compared to those without ODs, despite a noteworthy enhancement in memory and executive performance across all groups [64]. Persistent anosmia, even months after the infection, is also correlated with global cognitive dysfunction [63]. Clemente et al. [65] compared 32 post-COVID-19 patients with mild persistent hyposmia with a group of healthy controls. The investigation included Sniffin’ Sticks olfactory tests and cognitive assessments, along with EEG and functional near-infrared spectroscopy (fNIRS) during a Stroop test administered four months after infection. The results showed that post-COVID-19 subjects performed worse on the MoCA screening test and olfactory test and exhibited increased response latency in the Stroop test. These findings demonstrated that post-COVID-19 patients with persistent hyposmia exhibit mild prefrontal function deficits, with weaker prefrontal activation. Mental clouding, headaches, or a combination of the two symptoms were found to be strongly correlated with greater severity of olfactory impairments. Headaches and fogginess, in fact, could in turn reduce attention and concentration, reducing the accuracy of odor identification, suggesting a complex interconnection between olfactory and cognitive symptoms [66]. 

#### 3.3.2. Mood Disorders 

In the current scientific context, there is a growing interest in exploring the correlation between mood disorders (including anxiety, post-traumatic stress, and depression) and ODs post-SARS-CoV-2 infection. This trend reflects an increasing awareness of the need to thoroughly investigate the connections between neuropsychiatric manifestations and the consequences of SARS-CoV-2 infection, aiming to advance the understanding and treatment of such conditions [54,61,63,67]. The study by Dudine et al. [67] investigated the psychological effects of taste and smell dysfunctions in 104 subjects affected by or recovering from COVID-19. Participants underwent a semistructured interview regarding clinical symptoms, including taste and smell problems, and were assessed using the Distress Thermometer and the Hospital Anxiety and Depression Scale (HADS). The findings indicated that mild to moderate taste and smell dysfunctions are associated with higher levels of psychological distress. Similarly, Giordano Cecchetti’s work [64] explored cognitive impairments in 49 post-COVID-19 patients and found that the presence of hyposmia/anosmia during acute illness is associated with persistent depressive symptoms even 2 months after resolution, suggesting that sensory manifestations during infection may have lasting impacts on mental health. Faulet et al. [54] analyzed patients four months after hospitalization for acute COVID-19 infection. Those who had anosmia during acute infection also exhibited more pronounced depressive symptoms and PTSD. In contrast to the aforementioned studies, the study coordinated by Tania Llana [63] examining long COVID patients, despite demonstrating that these patients exhibit symptoms of anxiety and depression, did not find significant differences in mood disorders between groups with and without OD, suggesting complexity in the relationships between these variables.

### 3.4. Treatment Approaches for COVID-19-Related OD

The high incidence of hypo/anosmia resulting from COVID-19 has drawn attention to potential treatments for olfactory deficits. While spontaneous resolution occurs in most COVID-19-related cases of hypo/anosmia, therapeutic intervention may be considered if the loss persists beyond two weeks [68,69]. The efficacy of available treatments for COVID-19-related hypo/anosmia remains uncertain, encompassing pharmacological, supplement-based, and olfactory training-based therapies [69,70,71,72,73,74]. Research provides evidence of its effectiveness in enhancing olfactory function, especially with essential oils such as rose, lemon, clove, and eucalyptus, commonly selected according to the classification of primary odors, demonstrating a positive impact [75]. However, to integrate it into a clinical setting, a sustained and prolonged therapeutic intervention is necessary [76]. Other treatments, such as oral or intranasal corticosteroids, intranasal insulin, and integration with vitamin A, omega-3, zinc, and alpha-lipoic acid, have been explored, each with varying degrees of supporting evidence and associated risks [77,78,79,80,81,82,83,84,85,86,87,88,89,90,91,92,93,94]. There is a growing interest in evaluating how such approaches can be combined synergistically, aiming to maximize effectiveness in restoring olfactory function [95,96]. In particular, a combined therapy based on corticosteroids and olfactory training has shown potential benefits in some studies [73,97,98,99]. Other promising treatments for post-COVID OD include cerebrolysin and theophylline, each with distinct mechanisms and potential benefits [100,101,102,103,104]. Despite the array of therapeutic options, uncertainties persist regarding their overall effectiveness for post-COVID-19 OD, necessitating further research to elucidate their safety and efficacy [102].

## 4. Discussion and Future Directions

OD has emerged as a prominent symptom associated with COVID-19, and its impact extends beyond the acute phase of the infection. This review synthesizes the findings of magnetic resonance imaging (MRI) studies, highlighting morphological, volumetric, and functional alterations in the olfactory system among individuals experiencing persistent OD after SARS-CoV-2 infection. Additionally, we have investigated cognitive and mood deficits associated with post-COVID OD and possible available treatments. The MRI studies for COVID-19-related OD employed various methodologies, assessing structural and functional aspects of the olfactory system through various parameters [2,4,34,35,36,37,38,39,40,41,44,45,46]. Evaluation of OBV and the OS depth, common parameters in different studies, revealed a consistent reduction in OBV and OS depth in post-COVID-19 patients with OD, suggesting direct damage to olfactory neural pathways [2,35,37,38,39,40,44,46]. Only one, among the selected studies, the study by Abdou et al. [41], presented a new perspective, reporting an OBV and OB dimensions increase, emphasizing a potential variety in the pathophysiological mechanisms underlying COVID-related OD. A plausible hypothesis to explain the observed heterogeneity in study results may lie in the temporal dynamics of COVID-19 infection and its consequences on the olfactory system. It is possible that during an acute or post-acute phase of the infection, the OBV increases in response to inflammation, an immediate reaction mechanism of the central nervous system to viral aggression. However, persistent OD could reflect alterations in neurological structures, including the downregulation of olfactory receptors. Substantial damage to olfactory neurons could lead to progressive atrophy of OBs and other structures involved in the olfactory process, thus being responsible for the observed decrease in OBV and OS depth. In other words, while the initial phase of infection may be characterized by an increase in volume due to inflammation, the subsequent phase, marked by persistent neural damage, may manifest with progressive structural thinning [20]. If confirmed, this theory could contribute to explaining variations in OB measurements observed among different studies and underscores the importance of considering the time elapsed since infection in assessing the neurological conditions of COVID-19 patients with OD. The comprehensive overview of the literature provided emphasizes the multifaceted nature of OD in post-COVID-19 patients. A detailed understanding of the different radiological manifestations associated with COVID-19-related OD is essential in light of these results. We highlighted the growing interest in assessing post-COVID cognitive disorders through general screening methods and more in-depth neuropsychological evaluations [55,62,63,64,65,66]. Short- and long-term verbal memory emerged as one of the most affected cognitive domains, with significant correlations between olfactory discrimination and executive functions [62]. Consistent results were also found in other studies, where anosmia emerged as a reliable predictor of mnemonic performance [55,63,64]. The complex association between olfactory and cognitive symptoms becomes clear when considering that mental fog, headache, or their combination are strongly linked to the increased severity of olfactory alterations. These symptoms, in fact, could negatively impact attention and concentration, leading to decreased accuracy in odor identification and therefore poorer results in olfactory tests, as highlighted by Arianna di Stadio et al. [66]. Future research should delve into the underlying mechanisms connecting OD to cognitive decline and mood disorders, providing insight into potential therapeutic interventions and holistic care strategies for patients [3,61]. In this review, we also evaluated the therapies currently under study for COVID-19-related hypo/anosmia, and crucial considerations emerged outlining the complexity and challenges in managing this post-infectious condition. The therapeutic approach is diverse, ranging from pharmacological treatments such as corticosteroids and intranasal insulin to the intake of dietary supplements and strategies like olfactory training [82]. This diversity reflects the lack of a universal therapy and emphasizes the need to personalize interventions based on patient specificities. Current evidence suggests that managing post-COVID-19 hypo/anosmia through medications like corticosteroids requires careful consideration of the associated risks and benefits. The effectiveness of this treatment is uncertain, and this uncertainty is accompanied by concerns about potential risks associated with prolonged use or high dosages [93]. Particularly promising, however, is the olfactory training regimen, which has shown positive results in various clinical contexts [70,72]. Its effectiveness in improving olfactory function, especially in the domains of odor discrimination and identification, suggests a crucial role in managing post-COVID-19 hypo/anosmia [71,74,75]. In conclusion, despite the promises offered by some therapies, there remains a critical need for further large-scale clinical research to consolidate the validity of therapeutic options and delineate definitive guidelines for managing post-COVID-19 hypo/anosmia [105].

## 5. Limitations

While this review sheds light on the complex impact of COVID-19 on OD and associated cognitive disorders, it underscores several limitations across the current literature. Common constraints include small sample sizes, the absence of healthy control groups, and the use of cross-sectional designs, hindering the establishment of causation. Additionally, the lack of objective olfactory measurements and potential discrepancies in self-reported symptoms raise validity concerns. Overall, these limitations underscore the need to interpret the results with caution. While acknowledging that a systematic review could have mitigated some of the limitations we reported in our study and would have provided a more comprehensive and objective analysis, our review was intentionally designed to serve as a critical reflection on the constraints within the current literature on this topic. We aimed to emphasize the need for further research, with particular attention to larger samples, appropriate controls, objective measurements, and comparable examination timeframes.

## 6. Conclusions

In conclusion, in our review, we synthesized the MRI abnormality findings associated with COVID-19 infection. In particular, we analyzed the olfactory system’s morphological, volumetric, and functional alterations. Additionally, we investigated cognitive and mood deficits associated with post-COVID OD. The findings could provide valuable insights for predicting outcomes and exploring potential treatments for OD. Further research is essential to deepen the understanding of the long-term effects of COVID-19 on various aspects of health.

## Figures and Tables

**Figure 1 diagnostics-14-00359-f001:**
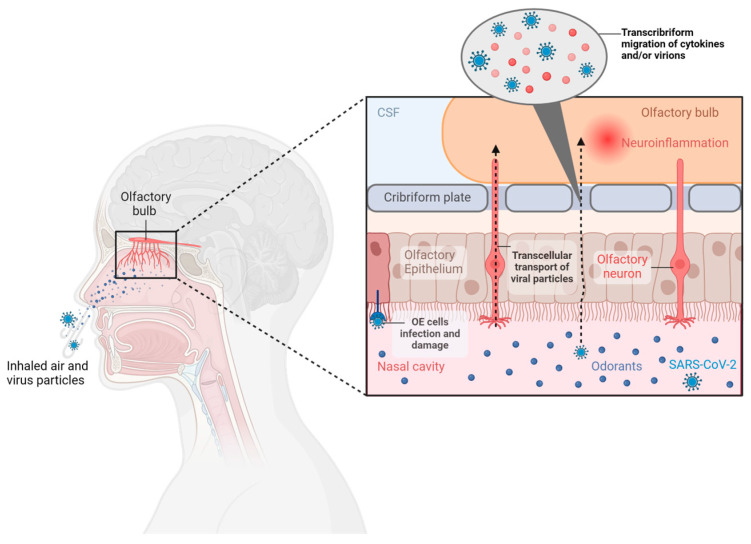
Anatomical illustration of the human olfactory system, with a specific focus on the OB and olfactory epithelium. The transformation of olfactory signals into electrical signals originates at the level of the olfactory epithelium. Subsequently, these signals are transmitted through olfactory neurons, constituting the fundamental interface for odor perception in the central nervous system to the OB. Additionally, the figure incorporates potential pathways through which SARS-CoV-2 may infect the olfactory bulbs, leading to inflammation. [The illustration was created using BioRender, https://www.biorender.com/ (accessed on 6 February 2024)].

**Figure 2 diagnostics-14-00359-f002:**
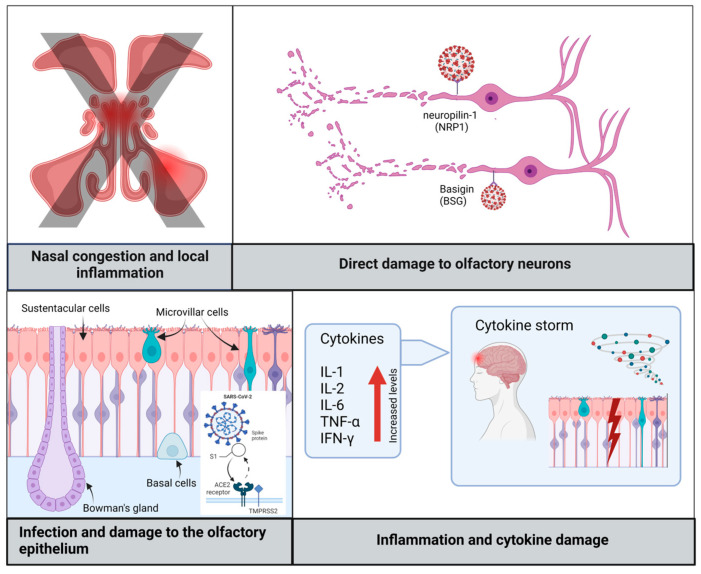
Comprehensive exploration of the pathogenesis of OD in SARS-CoV-2 infection. This illustration provides a detailed overview of the underlying pathogenic mechanisms of OD associated with SARS-CoV-2 infection. As evident, nasal congestion is systematically excluded as a predominant factor. The figure outlines intricate processes, including direct damage to neurons, alterations in supporting cells within the OE, and the role of inflammatory responses and cytokine damage. These findings contribute to a multifactorial understanding of the etiology of olfactory impairment in COVID-19. [The illustration was created using BioRender, https://www.biorender.com/ (accessed on 6 February 2024)].

**Figure 3 diagnostics-14-00359-f003:**
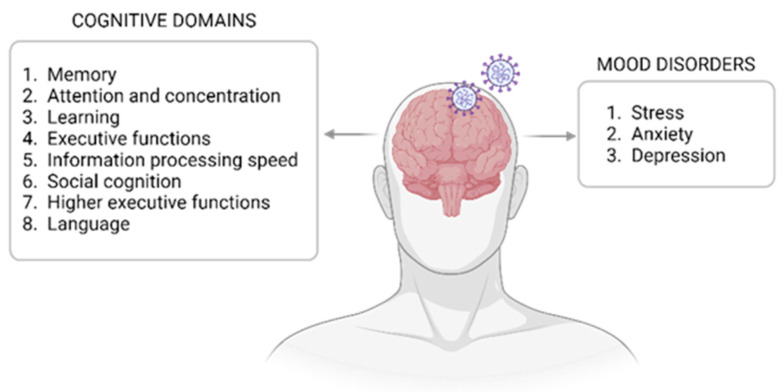
Interconnections between COVID, cognitive functions, and mood disorders. The figure presents a list of cognitive domains and mood disorders that have emerged as correlated with post-COVID ODs. Among cognitive domains, aspects such as memory and attention are included. Psychological disorders such as anxiety, stress, and depression are also represented, highlighting the connections between olfactory impairment and psychological impacts. This visual representation aims to underscore the complexity of interactions between ODs and associated cognitive and psychological phenomena. [The figure was created with BioRender, https://www.biorender.com/ (accessed on 6 February 2024)].

## Data Availability

Not applicable.

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
