# Peer review of "A Comprehensive Review of COVID-19-Related Olfactory Deficiency: Unraveling Associations with Neurocognitive Disorders and Magnetic Resonance Imaging Findings"

_diagnostics, 2024, doi:10.3390/diagnostics14040359_

Round 1

Reviewer 1 Report

Comments and Suggestions for Authors

This manuscript is a a Comprehensive Review of COVID-19 related Olfactory Deficiency with the aim of unraveling Associations with Neurocognitive Disorders and Magnetic Resonance Imaging Findings.

It is a dense review that needs to be reviewed for a better understanding to the reader.

I recommend its publication after some minor improvements:

·      Shortening the description of the structure of the olfactory system, and improving Figure 1 to better show possible viral targets.

·      Divide lines 165-223 in several paragraphs to help the reader to distinguish the different possible mechanisms.

·      Add an example of MRI findings in a COVID patient compared to a normal subject.

·      Add a paragraph on limitations of the study at the end of discussion, moving them from the conclusions.

·      In conclusions, remove any physiopathological discussion not addressed in this work.

Reviewer 2 Report

Comments and Suggestions for Authors

Simonini et al. conducted a review that aimed at examining the impact of COVID-19 on the olfactory system, focusing on MRI findings and the relationship between OD and neurocognitive disorders.

The review included 15 observational studies, case reports on MRI-detected structural changes in olfactory structures, and 10 studies on the correlation between smell loss and neurocognitive or mood disorders in COVID-19 patients (please specify how many case reports? Why was a systematic review not done?).

The findings show consistent MRI evidence of volumetric abnormalities and altered signal intensity in olfactory bulbs and cortex in patients with persistent OD. There's also a significant association between OD and cognitive impairment, memory deficits, and depressive symptoms (This has been already reported by other published reviews? I have major concerns about the novelty of this review).

The review acknowledged the limitations in existing literature, such as the lack of objective olfactory measurements and potential validity issues in self-reports, urging cautious interpretation (This could have been mitigated at least if you had conducted a systematic review). 

The review is well-conducted (as a narrative review) and well-written; however, a major concern arises regarding the novelty of the work.

Round 2

Reviewer 2 Report

Comments and Suggestions for Authors

Line 507: please change "limits" to "Limitations" and add the limitations related to the lack of systematic search and why a systematic nor scoping review was not conducted. 
